# Dietary Sources of Fructose and Its Association with Fatty Liver in Mexican Young Adults

**DOI:** 10.3390/nu11030522

**Published:** 2019-02-28

**Authors:** Alejandra Cantoral, Alejandra Contreras-Manzano, Lynda Luna-Villa, Carolina Batis, Ernesto A. Roldán-Valadez, Adrienne S. Ettinger, Adriana Mercado, Karen E. Peterson, Martha M Téllez-Rojo, Juan A. Rivera

**Affiliations:** 1Consejo Nacional de Ciencia y Tecnología (CONACYT), Mexico City 03940, Mexico; alejandra.cantoral@insp.mx (A.C.); carolina.batis@insp.mx (C.B.); 2National Institute of Public Health, Center for Research on Nutrition and Health, Cuernavaca, Morelos 62100, Mexico; lyluvi@yahoo.com.mx (L.L.-V.); adrianam@insp.mx (A.M.); mmtellez@insp.mx (M.M.T.-R.); jrivera@insp.mx (J.A.R.); 3Hospital General de Mexico “Dr. Eduardo Liceaga”, Directorate of Research, Mexico City 06720, Mexico; ernest.roldan@usa.net; 4Department of Radiology, Sechenov First Moscow State Medical University (Sechenov University), 119992 Moscow, Russia; 5Department of Nutritional Sciences, University of Michigan School of Public Health, Ann Arbor, MI 48109, USA; adrienne.ettinger@gmail.com (A.S.E.); karenep@umich.edu (K.E.P.)

**Keywords:** Fructose, High fructose corn syrup, NAFLD, hepatic index, Sugar-sweetened beverages

## Abstract

Fructose intake has been associated with non-alcoholic fatty liver disease (NAFLD). The objective of this study was to assess the consumption of dietary fructose according to: 1) classification of hepatic steatosis by two indexes and 2) diagnosis of NAFLD by MRI. We conducted a cross-sectional analysis among 100 young adults from Mexico City. The Hepatic Steatosis Index (HSI) and the Fatty Liver Index (FLI) were estimated using Body Mass Index (BMI), waist circumference, and fasting concentrations of glucose, triglycerides, and hepatic enzymes (ALT, AST, GGT). A semi-quantitative food frequency questionnaire was administered to obtain dietary sources of fructose. We estimated the concordance between the hepatic indices and NAFLD and the correlation between the index scores and the percentage of liver fat. Eighteen percent presented NAFLD; 44% and 46% were classified with hepatic steatosis according to HSI and FLI, respectively. We compared dietary intake of fructose by each outcome: HSI, FLI, and NAFLD. Sugar-sweetened beverages (SSB) and juices were consumed significantly more by those with steatosis by FLI and NAFLD suggesting that SSB intake is linked to metabolic alterations that predict the risk of having NAFLD at a young age.

## 1. Introduction

Non-alcoholic fatty liver disease (NAFLD) is defined as the excessive accumulation of triglycerides (>5%) in the hepatocytes in the absence of significant alcohol consumption [1,2]. This hepatic steatosis can progress to fibrosis and cirrhosis [3,4]. It is commonly found in overweight individuals, and is associated with dyslipidemia, visceral adiposity, hypertension, and insulin resistance [5,6]. Patients with NAFLD have increased risk for hepatic malignancy and cardiovascular events [7]. NAFLD has been increasing in the last 20 years in adolescents and adults in the U.S. [8,9]. In Mexico, the national prevalence of NAFLD is unknown, but has been estimated to be between 20 to 29.9% [10,11].

The assessment of hepatic steatosis requires liver biopsy or a radiologic measure such as magnetic resonance imaging (MRI), both not suitable for diagnosis at the population level. There is no consensus for a noninvasive reference to predict NAFLD for screening at population level, but several predictive models based on biochemical data have been developed as a tool for clinicians to identify patients at risk for further diagnostic evaluation and intervention, which may be indicated at early stages. The Fatty Liver Index (FLI) and the Hepatic Steatosis Index (HSI) have been validated showing a medium to high sensitivity, but they cannot quantify the severity of hepatic steatosis [12,13].

Behavioral factors are involved in the pathophysiology of fatty liver, such as sedentary lifestyle and dietary factors: diets high in total energy intake, saturated fats, and fructose [14,15]. Fructose is a monosaccharide present in natural foods like fruits and honey, but also is the major component of two sweeteners commonly used as food additives: sucrose (disaccharide composed by fructose and glucose) and high fructose corn syrup (HFCS) (composed by the solution of two free monosaccharide: fructose and glucose) [16].

Many processed food products contain HFCS since its introduction as a sweetener in the 1960s, due to the advantage to food manufacturers in providing better flavor, stability, freshness, texture, color, and consistency in comparison to sucrose [16]. Sugar sweetened beverages (SSB) such as soft drinks and other carbohydrate-sweetened beverages are usually composed of 55% fructose [16].

Previous studies have shown that consumption of fructose, through soft drinks and other beverages, is higher in NAFLD patients than in controls [17,18]. Mexico has one of the highest per capita intake of soft drinks worldwide and their consumption is particularly high in the 19 to 29 year-old age group [19]. In addition, 26% of total calories in the Mexican diet come from discretionary foods (processed foods with high energy and low nutrient density), including SSB, pastries, and high energy dense snacks [20]; SSB contribute to about 69% of added sugars, and 13% of total energy intake to the Mexican diet. [21]

The aim of this study was to assess the dietary consumption of fructose according to: the classification of hepatic steatosis by the FLI and HSI, and the diagnosis of NAFLD by MRI in Mexican young adults.

## 2. Materials and Methods

The project protocol was reviewed and approved by the Ethics, Biosafety, and Research Committees of National Institute of Public Health, Mexico.

### 2.1. Study Population

A cross-sectional analysis was performed in a sample of 100 healthy young adults living in Mexico City between October of 2016 and May of 2017. Participants were selected from the Early Life Exposure in Mexico to Environmental Toxicants (ELEMENT) cohort study which has been described elsewhere [22]. In summary, between 1994 and 1995, mothers were recruited at delivery, and their off-spring were followed during early childhood. Information from pregnancy, delivery and post-partum up to 48 months of age was obtained. A total of 631 mother–child participants encompass the original cohort. From 2016 to 2017, a subsample of 100 offspring participants were invited to participate in this current study and had complete information for the analysis. Comparison of the original cohort and the analytical sample is presented in Appendix A.

For the present study, participants were evaluated during a weekend day at the research center after 10 hours of fasting. Once at the center, they were informed about the objectives and risks of the study and signed a letter of informed consent. A blood sample and anthropometric measures were obtained. In order to estimate the hepatic triglyceride content, proton magnetic resonance spectroscopy (PMRS) was performed. A trained nutritionist administered validated questionnaires: a semi-quantitative food frequency questionnaire (FFQ) and a lifestyle questionnaire.

#### 2.1.1. Magnetic Resonance Imaging (MRI)

Measurement of liver fat content and diagnosis of NAFLD was made by MRI. In order to estimate the hepatic triglyceride content, we performed PMRS [23], with the calculation of proton density fat fraction (PDFF). A Philips Achieva 3.0 T MR-scanner (Philips Healthcare, Best, The Netherlands) was used for imaging. The resonances were used for calculation of the triglycerides were water (peak at 4.7 ppm), methylene (peak at 1.3 ppm, [CH2]n) and methyl (peak at 0.9 ppm, CH3). A detailed description of the imaging examination and post-processing analysis has been described [24,25]. An example of the acquired images and spectra is depicted in (Appendix A). When hepatic triglyceride content is measured with MRI proton density fat fraction, a cut-off value of 5% has been used to define NAFLD [26].

#### 2.1.2. Hepatic Indices

A fasting blood sample was obtained to quantify glucose, triglycerides and hepatic enzymes (ALT, AST, GGT) using a bench clinical chemistry analyzer (Diasys response 910, Diagnostic Systems GmbH, Holzheim, Germany).

Weight and height were measured using a Tanita digital scale with height road (Model WB-3000m, Tanita Corporation, Tokyo, Japan). Weight was recorded to the nearest 0.1 kg and height to the nearest 0.5 cm. Body Mass Index (BMI) was calculated using both measures. Waist circumference (WC) was measured twice to the nearest 0.1 cm with a SECA measuring tape (Model 201, Seca, Hamburg, Germany) in each participant and the average of both measures was included in the analysis. All the measurements were obtained by trained personnel.

The Hepatic Steatosis Index (HSI) was estimated using: the ALT/AST ratio, BMI, sex, and impaired fasting glucose (IFG) blood levels (>110 mg/dL) [27], instead of the diagnosis of diabetes mellitus (DM) as this is a young and “healthy” sample. The formula used was:HSI = 8 × ALT/AST + BMI (+2, if IFG;+2, if female)

Once the index was obtained, the rule of ≥36 points was used to classified an individual with hepatic steatosis with a sensitivity of 92% [13].

In the case of the Fatty Liver Index (FLI), triglycerides, BMI, GGT, and WC were used to obtain the score. The formula used was:$$\mathrm{FLI}\;=\;\frac{e\;0.953*\mathrm{loge}\;\left(\mathrm{triglycerides}\right)\;+\;0.139*\mathrm{BMI}\;+\;0.718*\mathrm{loge}\;\left(\mathrm{ggt}\right)\;+0.053*\mathrm{WC}\;-\;15.745}{1\;+\;e\;0.953*\mathrm{loge}\;\left(\mathrm{triglycerides}\right)\;+\;0.139*\mathrm{BMI}\;+\;0.718*\mathrm{loge}\;\left(\mathrm{GGT}\right)\;+\;0.053*\mathrm{WC}\;-\;15.745}\;*\;100$$

Once the score was estimated, having ≥30 was use to classified the participants with hepatic steatosis with a sensitivity of 87% [12].

#### 2.1.3. Dietary Information

Dietary data was collected electronically using a validated, semi-quantitative FFQ based on the one used in the National Health and Nutrition Survey (ENSANUT-2012) which included 140 food and beverages classified into 14 food groups [28]. The data collected included the number of days, times per day, serving size, and number of servings consumed of each food and drink listed, during the seven days prior to the interview.

To process the dietary information, the quantity of each food and drink was obtained by multiplying the number of days by the times per day, by the portion size (grams (g), millimeters (mL)) and by the number of portions or pieces consumed on each occasion. Total g and mL were divided by seven days to obtain the average daily intake. For each food consumed, energy and macronutrients were calculated using a nutritional composition database of foods compiled by the National Institute of Public Health [29].

Dietary sources of fructose were classified into five food groups from the foods in the FFQ and their commonly consumed portions, as follows:

(1) SSB intake (240 mL/day). Defined as the sum of mL per day of: soda, sugar-sweetened commercial fruit beverages, sugar-sweetened commercial tea or flavored water beverages, home-made fruit beverages (“aguas frescas” commonly consumed in Mexico) with sugar, coffee with sugar, and tea with sugar.

(2) Cereals, bars, and bread (100 g/day) was equal to the sum (g) of cereal bars, breakfast cereals, cookies, sweet bread, and pastries.

(3) Candies (10 g/day)

(4) Natural juice (no added sugar) consumed per day (240 mL/day)

(5) Fruits consumed per day (200 g/day)

Frequency of the intake of commercial foods and beverages that reported fructose as an ingredient in their labels was also identified: “Fructose,” “High fructose corn syrup,” or “Fructose corn syrup.” In 2015, a list of the most available commercial foods was developed by our research group in supermarkets, convenience stores, and grocery stores in Mexico City with the intention of classifying commercial foods according to the type of sweetener (data not published). This facilitated the identification by the participants of the products, brands, and types of foods which contain added fructose. Also, participants were shown pictures of the labels of each product listed. Additionally, if the subject reported any brand other than from the predetermined list, the information was captured and included in the consumption of fructose if the product contained it.

#### 2.1.4. Other Variables

Information about sex, age, socioeconomic status (SES), physical activity, and alcohol habits were collected. For the SES, the AMAI algorithm (Asociación Mexicana de Agencias de Investigación de Mercado) was used to obtain a six- category classification [30]. However, because the sample did not have a wide variability in SES, we collapsed the six categories into two (low and middle SES). The short version of the International Physical Activity Questionnaire was applied to obtain the categories of physical activity (inactive, minimal active, active) [31]. Questions about alcohol intake were taken from the National Addiction Survey and intakes <20 g/day of alcohol was consider as non-related to fatty liver disease, which correspond to less than 7 and 14 drinks per week for women and men, respectively [32].

#### 2.1.5. Statistical Analysis

Sociodemographic and lifestyle characteristics are presented in means (SD) and proportions (n) for the entire sample and according to both indexes and the diagnosis of NAFLD by MRI, variables were compared using t-test or Fisher’s exact test, depending on the nature of the variable. Dietary variables are presented as medians (Interquartile range: IQR) due to their skewed distribution.

As a first step, we estimated the sensitivity and specificity of the indexes. We identified the “true positives” as those classified as having steatosis by the index (HSI ≥ 36 or FLI ≥ 30) and also as NAFLD by the MRI (>5% of triglyceride content in the hepatocytes). We identified “false positives” as those classified as having steatosis by the index (HSI ≥ 36 or FLI ≥ 30), but not with NAFLD by the MRI (≤5% of triglyceride content in the hepatocytes). We also estimated the Spearman correlation between each index score and the percentage of liver fat (by MRI).

Multinomial logistic regressions were fit to compare all risk factors measured (BMI, Waist Circumference, ALT, AST, GGT, triglycerides, glucose, and subcutaneous and visceral fat) between healthy subjects (<36 points in HSI and <30 points in FLI and ≤5% of fatty liver by MRI) and those classified as true positive and false positive. We aimed to do this estimation as a way to show that even those identified as false positive may have an increasing risk profile in the parameters measured comparing to those categorized as healthy. We also compared true positive versus false positive subjects. All models were adjusted by: sex, smoke habit, physical activity category, alcohol intake category, energy intake and BMI. Models for Waist Circumference and BMI were not adjusted by BMI. To identify the differences in the consumption of each food group’s sources of dietary fructose (SSB; cereals bars and bread; candies; fruits and natural juice) and the classification of the FLI, and HSI (categorized as Positive, Negative or False Positive), and according to NAFLD diagnosis by MRI, we used Kolmogorov Smirnov Test.

All calculations were completed using Stata 13.0 (StataCorp LLC, College Station, TX, USA), and a statistical significant value of α < 0.05 was considered.

## 3. Results

According to the MRI, 18% of the participants had the diagnosis of NAFLD (>5% of triglyceride content in the hepatocytes), and according to HSI and FLI, 44% and 46% of the participants presented hepatic steatosis, respectively. Table 1 presents sociodemographic and lifestyle characteristics of the participants. This is a sample with a mean age of 21.4 (+0.5) years, 54% were male, and 52% were classified as low SES. Forty-five percent reported to be active smokers and 41% past smokers. Alcohol intake was null or minimal in 48%, while 20% reported drinking > 4 drinks weekly. None of the participants reported alcohol intake higher than 20 g/day of alcohol. Fifty-six percent of participants were not physically active. There were no statistically significant differences among these variables. Only smoking habit (never, past smoker, active smoker) was significantly different among participants with hepatic steatosis (FLI) or NAFLD.

Figure 1 presents the concordance between percentage of fat in the liver measured by MRI and the two indexes (FLI and HSI). The correlation between the score of both indexes was positive and statistically significant (*r* = 0.62 FLI, *r* = 0.60 HSI, *p* < 0.01). Using the cut-point of 36 points for HSI and 30 for FLI the sensitivity of both indexes to predict steatosis was 77.8%; meanwhile the specificity was 63.4% and 61 % for HSI and FLI, respectively.

Table 2 presents the association between metabolic risk factors for NAFLD, HSI, and FLI versus the healthy participants. As we expected, the comparison between the true positive participants and the healthy participants showed that the first ones had an elevated risk in almost all the parameters in a higher level (including the parameters used to calculate the index but also the ones not used for the construction of their respective index). When comparing those classified as false positive versus those in the healthy category, we identified that BMI is 2.3 to 3 times higher in the false positive group according to the HSI and FLI, respectively. For example: in the case of the 30 participants with a HSI ≥ 36 but with <5% of fatty liver by MRI (false positives) the risk is significantly higher in the variables waist circumference (RRR = 1.41, 95%CI 1.09–1.83) and GGT (RRR = 1.16, 95%CI 1.05–1.29) compared to the healthy participants, these two parameters are not consider in the HSI estimation. For the false positive participants in the FLI the risk is significantly higher in the variable ALT (RRR = 1.10, 95%CI 1.02–1.20) compared to healthy participants, also ALT is not consider in the estimation of the FLI index. So, besides the sensitivity of the indexes to predict steatosis, both indexes reflect a population with a metabolic risk profile.

Table 3 presents the dietary information for the entire sample and by categories: Healthy, HSI, FLI, and NAFLD. For the complete sample (*n* = 100), dietary information showed that the median (IQR) energy intake was 2689 (1682) Kcal with approximately 56% of the calories from carbohydrates, 33% from lipids, and 13% from proteins. Median (IQR) intake of SSB was 720 (1037) mL with the main contributor to SSB being soda (specifically cola-type) which contributed 45% of the total SSB, followed by sugar-sweetened commercial fruit beverages and home-made fruit beverages with 22%. The median intake of SSB with added HFCS was 9 mL; however, the range of intake was large (IQR 107 mL). The consumption of cereals, bars, and bread products was 91 (89) g with the intake of those with added HFCS equal to 10 (26) g. The median consumption of natural fruit juices was almost null and the median (IQR) fruit intake was 258 (335) g per day. Even we explored the intake of diet beverages, only 3 participants reported to consumed diet soda the previous week.

When comparing the dietary information by FLI classification, it is important to note that those with negative FLI (score < 30) consumed statistically less energy (kcal) and total grams of carbohydrates per day, but also reported consuming more calories from proteins and lipids as a percentage of the total calories, compared to those classified as positive FLI (score ≥ 30) and also compared to those classified as false positives. In the case of the HSI classification, we found the same pattern but it was not statistically significant (Table 3).

The dietary intake of almost all dietary sources of fructose were higher in those classified with steatosis by both indexes, but there was no significantly different, except for the SSB intake according FLI: 536 (837) mL in the group classified as not having steatosis (score < 30), 960 (959) mL in the group classified as having hepatic steatosis (score ≥ 30), and 1273 (1539) mL in those classified as false positive (*p*-value = 0.04). When we compare the consumption of the different beverages included in the SSB, the soda intake was statistically higher in FLI false positive than in negative subjects. In the case of those classified as NAFLD, the median intake of natural fruit juices was statistically higher compared to those classified as non-NAFLD (146 versus 0 mL/day).

## 4. Discussion

In this study, we compared the intakes of different dietary sources of fructose in relation to two liver indexes that predict hepatic steatosis and in relation to the identification of NAFLD by MRI in young adults in Mexico. First, we found that 18% of the participants presented NAFLD by MRI, which is similar to previous reports in this age group [33]. Regarding the hepatic indexes, we found that, in our sample, the sensitivity was around 77%, which is also consistent with previous publications [34,35], and the prevalence of steatosis (44% and 46% according to HSI and FLI, respectively) was also similar to a previous study in Mexico (49%) [36]. We evaluated the risk profile of those classified as true positive and false positive compared to those identified as healthy and found that even those in the false positive group had higher levels of the metabolic risk parameters indicating that the indexes can capture population at risk of NAFLD.

We found that those participants classified as positive by FLI (true positive and false positive) consumed more SSB than those classified as negative, and that those classified as NAFLD consumed more natural juices than those without the diagnosed. We did not find differences in the intake of the other sources of fructose. These findings are consistent with previous studies that have found an association between dietary intake of soft drinks and sweetened beverages with NAFLD [17,18]. SSB can have sucrose or HFCS as added sweeteners and it has also been shown that fructose and sucrose alter hepatic insulin sensitivity and lipid metabolism when compared to glucose [37]. We did not expect to find a lower report of SSB intake in those classified as NAFLD compared to FLI and HSI, this could be explained by an under-report related to BMI as 94% of those identified with NAFLD were overweight, previous reports have indicated that overweight persons tend to underreport food items due to negative health image [38,39,40].

The main source of SSB in this population was soda. It is recognized that soft drinks can increase liver fat accumulation by different paths: rapid absorption of carbohydrates, increase glucose and insulin concentration in blood after its intake, rapidly metabolized by human liver [41,42]. In addition, cola-type caramel is rich in advanced glycation end products increasing insulin resistance and inflammation [43,44].

Fructose metabolism differs from glucose metabolism, after its absorption the majority (90%) of the fructose is cleared by the small intestine, but at high fructose amounts (>1 g/kg) it spills to the liver [45]. Within the liver, fructose is phosphorylated via fructokinase (not regulated by energy intake) and converted into trioses that can be converted to glucose (gluconeogenesis) or to generate products like triglycerides [46,47]. Intervention trials have provided evidence that high fructose leads to increases in de novo lipogenesis, blood triglycerides, and hepatic insulin resistance [48,49,50]. Additionally, fructose load does not suppress ghrelin and does not stimulate insulin or leptin [51]. For centuries, diets contained small amounts of natural fructose from fruits, but the increasing consumption of fructose as an added sweetener affects the high glucose/low fructose diet which is the one that humans are adapted [41]. Therefore, the recent high flux of fructose to the liver alters the hepatic carbohydrate metabolism potentially leading to the novo lipogenesis [52].

The fact that we did not find association of other food groups sources of fructose and NAFLD could be explained by the fact that those food groups, such as fruits and cereals, have other nutrients, fiber, and dietetic components that can alter the absorption of fructose (from sucrose or from HFCS). We also found that products with HFCS, with the exception of candy, are consumed at low levels in this population sample.

The limited sample size and the cross-sectional nature of this analysis limits our ability to draw conclusions regarding increased fructose consumption, mainly through SSB, on the natural history or progression of NAFLD in this population. Even though, a similar study with small sample size evaluated the association FLI and fructose intake (total fructose intake, fructose intake from fruits, fructose intake from juices, fructose intake from SSB) showing also significant results for SSB consumption [53]. A main strength of this analysis is our ability to compare results derived from two hepatic indexes to the gold standard diagnosis of NAFLD by MRI.

Dietary components are modifiable risk factors for metabolic and liver diseases that could be addressed in the early stages of life as an opportunity to prevent the progression of liver damage and other negative consequences. In particular, there is clear evidence of the negative effect of SSB intake, at the levels consumed by the Mexican population, on obesity, type 2 diabetes, cardiometabolic conditions, and dental caries [54,55,56,57]. Previous randomized controlled trials have found that obese subjects are more sensitive to present increases in fatty liver due to the intake of glucose- or fructose-sweetened beverages at isocaloric diets in comparison with normal weight subjects [58]. According our results, almost 100% of subjects diagnosed with NAFDL presented overweight or obesity. Additionally, we also observed that FLI subjects had on average a higher consumption of total energy and SSB, but not a higher percentage of calories derived from CHO (58.7% versus 54% negative FLI), this finding suggest that the relations go further the amount of calories, but more in the type of diet components.

This study adds to the existing evidence linking SBB intake with NAFLD, a disease that is increasing in Mexico [10]. Therefore, more public health action is needed to reduce the intake of fructose, as sucrose or HFCS, especially from SSB that are highly consumed in Mexico and in other populations. As obesity is currently one of the biggest health problems in Mexico, and it is related to the incidence of NAFLD [59,60], more studies are needed to measure the prevalence of NAFLD in relation to dietary components.

## Figures and Tables

**Figure 1 nutrients-11-00522-f001:**
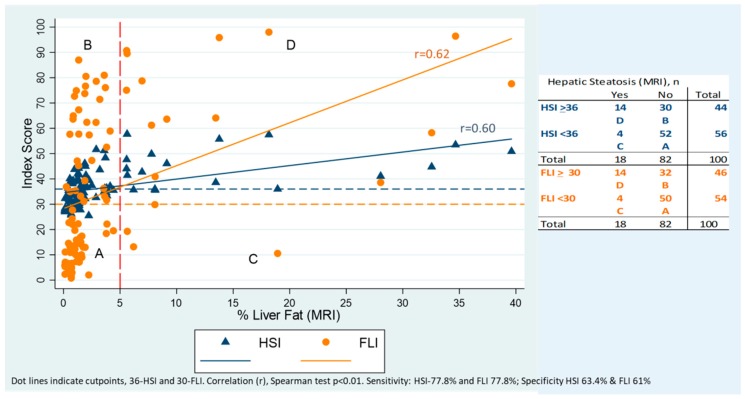
Correlation between the Hepatic Steatosis Index (HSI) and Fatty Liver Index (FLI) with the amount of fat in the liver (%) by MRI.

**Table 1 nutrients-11-00522-t001:** Sociodemographic and lifestyle characteristics of the participants in the entire sample and according to the Indices classification and the diagnosed of non-alcoholic fatty liver disease (NAFLD).

		HSI		FLI		NAFLD	
	All	<36 points	>=36 points	<30 points	>=30 points	<5% of liver fat content	>=5% of liver fat content
	*n* = 100	*n* = 56	*n* = 44	*n* = 54	*n* = 46	*n* = 82	*n* = 18
	*n* (%)	*n* (%)	*n* (%)	*n* (%)	*n* (%)	*n* (%)	*n* (%)
Age, years	21.4 (0.5)	21.5 (0.5)	21.4 (0.5)	21.5 (0.5)	21.4 (0.5)	21.5 (0.5)	21.4 (0.5)
Sex, male (%)	54	29 (54)	25 (46)	25 (46)	29 (54)	25 (46)	29 (54)
Marital status							
Single	80	47 (59)	33 (41)	44 (55)	36 (45)	44 (55)	36 (45)
Married/cohabiting	20	9 (45)	11 (55)	10 (50)	10 (50)	10 (50)	10 (50)
Occupation							
Student	33	19 (58)	14 (42)	17 (51)	16(49)	17 (51)	16(49)
Employee/home/Other	67	37 (55)	30 (45)	37 (55)	30 (45)	37 (55)	30 (45)
Socio economic status							
Low	52	22 (56)	23 (44)	28 (54)	24 (46)	28 (54)	24 (46)
Meddium	48	27 (56)	21 (44)	26 (54)	22 (46)	26 (54)	22 (46)
Smoke cigarettes							
Never	14	8 (57)	6 (43)	**9 (64)**	**5 (36)**	**14 (100)**	**0 (0)**
Past smoker	41	26 (63)	15 (37)	**27 (66)**	**14 (34)**	**34 (82.9)**	**7 (17.1)**
Active smoker	45	22 (49)	23 (51)	**10 (40)**	**27 (60)**	**34 (75.6)**	**11 (24.4)**
Alcohol intake							
Never/Annualy	48	25 (52)	23 (48)	28 (58)	20 (42)	39 (81.2)	9 (18.8)
Monthly	32	20 (63)	12 (37)	17 (53)	15 (47)	24 (75)	8 (25)
Weekly	20	11 (55)	9 (45)	9 (45)	11 (55)	19 (95)	1 (5)
Physical Activity							
Inactive	18	8 (44)	10 (56)	7 (39)	11 (61)	15 (83.3)	3 (16.7)
Minimal active	38	24 (63)	14 (37)	25 (66)	13 (24)	31 (81.6)	7 (18.4)
Active	44	22 (55)	20 (45)	22 (50)	22 (50)	36 (82)	18 (18)

Bold indicates statistically significant difference; *p* < 0.05 (*t* test or Fisher’s test).

**Table 2 nutrients-11-00522-t002:** Association between metabolic risk factors for healthy subjects, and those with NAFLD, HSI, and FLI (*n* = 100).

**Risk Factors Considered in the Index**	**Subgroup of Study and Reference**	**False Positive HSI Or at Risk versus Healthy**	**True Positive HSI versus Healthy**	**True Positive HSI versus False Positive HSI**	**False Positive FLI Or at Risk versus Healthy**	**True Positive FLI versus Healthy**	**True Positive FLI versus False Positive FLI**
	**Risk factor**	**RRR (CI 95%)**	**RRR (CI 95%)**	**RRR (CI 95%)**	**RRR (CI 95%)**	**RRR (CI 95%)**	**RRR (CI 95%)**
HSI and FLI	BMI	3.01 (1.74, 5.21) ^£^	3.64 (2.03, 6.54) ^£^	1.21 (0.96, 1.52)	2.35 (1.55, 3.54) ^£^	2.80 (1.78, 4.39) ^£^	1.19 (0.96, 1.47)
FLI	Waist Circumference (cm)	1.41 (1.09, 1.83) ^£^	1.34 (1.01, 1.79) ^£^	0.95 (0.80, 1.13)	1.27 (1.02, 1.59) ^£^	1.12 (0.87, 1.45)	0.88 (0.74, 1.06)
HSI	ALT (U/L)	1.20 (1.06, 1.35) ^£^	1.32 (1.15, 1.52) ^£^	1.10 (1.02, 1.19) ^£^	1.10 (1.02, 1.20) ^£^	1.21 (1.09, 1.35) ^£^	1.10 (1.03, 1.18) ^£^
HSI	AST (U/L)	1.11 (0.98, 1.27)	1.20 (1.05, 1.38) ^£^	1.08 (1.01, 1.16) ^£^	1.10 (0.97, 1.23)	1.20 (1.06, 1.37) ^£^	1.10 (1.02, 1.18) ^£^
FLI	GGT (U/L)	1.16 (1.05, 1.29) ^£^	1.16 (1.04, 1.29) ^£^	0.98 (0.95, 1.04)	1.22 (1.09, 1.37) ^£^	1.22 (1.08, 1.38) ^£^	0.99 (0.96, 1.04)
FLI	Triglicerydes (mg/dL)	1.00 (0.99, 1.01)	1.02 (1.00, 1.03) ^£^	1.01 (1.00, 1.03) ^£^	1.13 (1.02, 1.25) ^£^	1.14 (1.04, 1.26) ^£^	1.01 (0.99, 1.03)
HSI	Glucose (mg/dL)	1.03 (0.96, 1.11)	1.14 (1.01, 1.30) ^£^	1.11 (1.00, 1.23) ^£^	1.03 (0.95, 1.10)	1.14 (1.01, 1.28) ^£^	1.11 (1.00, 1.13) ^£^
**Other Risk Factors for NAFLD**							
	Subcutaneous fat (cm^2^)	1.02 (1.00, 1.04) ^£^	1.01 (0.99, 1.03)	0.99 (0.98, 1.00)	1.02 (0.99, 1.03)	1.00 (0.98, 1.03)	0.98 (0.98, 1.00)
	Visceral fat (cm^2^)	1.00 (0.96, 1.04)	1.03 (0.98, 1.08)	1.03 (0.98, 1.07)	0.99 (0.95, 1.03)	1.01 (0.96, 1.06)	1.02 (0.98, 1.05)

RRR; Relative Risk Reduction. Significantly different to healthy ^£^: *p* < 0.05; Healthy (*n* = 56), False Positive HSI (*n* = 30), True Positive HSI (*n* = 14), False Positive FLI (*n* = 32) and True Positive FLI (*n* = 14). Bold indicates statistically significant difference.

**Table 3 nutrients-11-00522-t003:** Daily dietary information in the complete sample and according to the Indexes classification, median (IQR).

		Healthy	HSI			FLI			NAFLD	
					False Positive			False Positive		
	All		<36 points	>=36 points	>36 points + no-NAFDL	<30 points	>=30 points	>=30 points + no-NAFLD	<5% of liver fat	>=5% of liver fat
	*n* = 100	*n* = 46	*n* = 56	*n* = 44	*n* = 30	*n* = 54	*n* = 46	*n* = 32	*n* = 82	*n* = 18
Energy, Kcal/day	2689 (1682)	2430 (2083)	2557 (1768)	2801 (1482)	2867 (1687)	**2380 (1885)**	**2953 (1305)**	**3020 (1518)**	2801 (1966)	2363 (1471)
Proteins, g/day	85.7 (54.5)	84.9 (66.6)	85.7 (49.6)	86.2 (56.4)	91.6 (55.4)	82.4 (46.6)	91.6 (57.5)	93.4 (48.7)	88.1 (57.5)	81 (34.4)
Calories from proteins (%)	13 (2.8)	13.3 (3.6)	13.3 (2.9)	12.7 (3)	12.5 (2.8)	**13.3 (3)**	**12.6 (2.6)**	**12.5 (2.7)**	13.1 (2.8)	12.8 (2.5)
Carbohydrates (CHO), g/day	375 (268.8)	338.1 (294.8)	347.9 (279.2)	419 (237.8)	435 (263.9)	**338.1 (298.8)**	**442.9 (194.6)**	**453.6 (198.5)**	375.7 (272.8)	363.6 (238.1)
Calories from CHO (%)	55.8 (12.4)	53.8 (11.4)	54.6 (12.9)	58.5 (11.3)	57.2 (11)	54 (10.9)	58.7 (13.8)	58.7 (13.3)	54.6 (13.3)	58.9 (9.9)
Sugar (g/day)	25.8 (31.9)	24.1 (24.8)	24.6 (28.8)	28.7 (33.1)	26.9 (33.3)	23.6 (25.5)	29.9 (29.8)	28.3 (37.4)	25 (32.7)	31.9 (28.5)
Calories from sugar (%)	3.9 (3.9)	3.8 (3.4)	3.8 (3.4)	4.1 (4.2)	4.1 (4.2)	3.7 (3.4)	4.2 (4.1)	4.2 (4.1)	3.9 (3.5)	5 (4.9)
Total fiber, g/day	26.1 (26.1)	25.3 (29)	25.3 (24.7)	28 (26.2)	25 (25.8)	25.3 (29.5)	27.1 (24.6)	24.5 (21.5)	24.8 (25.8)	34.4 (25)
Lipids, g/day	98.4 (66.4)	98.6 (79.2)	100.5 (66.9)	94.2 (66.9)	100.9 (66.7)	90.9 (69)	102.9 (64.3)	108.6 (60.9)	100.5 (70.3)	80.1 (55.6)
Calories from lipids (%)	33.1 (10.3)	36.1 (9.8)	34.5 (10)	32.3 (10)	32.4 (9.8)	**35.3 (9.3)**	**30.4 (11.2)**	**30.3 (10.4)**	34.2 (10.2)	30.5 (7.5)
Fructose Food Groups										
Sugar-sweetened beverages (SSB) (mL/day)	**720 (1037.1)**	**518.6 (874.3)**	557.1 (921.4)	895.7 (977.1)	1045 (1440)	**535.7 (857.1)**	**960 (959.9)**	**1272.9 (1538.6)**	720 (1105.7)	690.0 (565.7)
SSB with HFCS (mL/day)	9 (107.2)	9 (85.8)	8.9 (94.3)	16.9 (174.3)	0 (107.2)	0 (85.8)	35 (177)	52.2 (142.1)	8.9 (107.1)	17.1 (191.4)
Cereals, bars and bread (g/day)	90.7 (88.9)	105.9 (141.7)	96.2 (128.9)	97.0 (97.5)	109.32 (85)	81.4 (126.3)	109.3 (84.3)	111.5 (67)	107.2 (112.5)	76.2 (116.3)
Cereals, bars and bread with HFCS (g/day)	10 (25.8)	9.2 (32.1)	12.2 (30.9)	8.9 (24.9)	9 (25.7)	8 (25.7)	12.5 (29.6)	16 (28.3)	11.3 (25.9)	8.9 (18.2)
Candies (g/day)	6 (16.2)	2.9 (14.5)	6.8 (17.6)	5.3 (14.7)	5.3 (13.8)	2.4 (14.5)	7.8 (23.3)	7.8 (18.2)	5.3 (15.2)	8.2 (23.6)
Candies with HFCS (g/day)	8.6 (22.9)	11.1 (25.8)	11.1 (25.4)	5.8 (19)	4.7 (19.7)	9.7 (22.2)	8.6 (20)	7.2 (20.2)	8.6 (21.5)	9.3 (28.6)
Natural fruit juices (mL/day)	0 (102.9)	0 (102.9)	0 (102.9)	0 (85.8)	0 (0)	0 (102.9)	0 (102.9)	0 (68.6)	**0 (68.6)**	**145.8 (377.2)**
Fruit (g/day)	258 (335.4)	223.6 (298.8)	247.8 (297.3)	415.8 (415.8)	255.8 (349.1)	230.9 (325.1)	306.9 (390.6)	281.3 (319.2)	238.6 (318)	383.2 (549.3)
**SSB (ml) Detail**										
Coffe with sugar (mL/day)	0 (102.8)	0 (102.8)	0 (102.8)	0 (171.4)	0 (240)	0 (102.8)	0 (235.7)	0 (246.4)	0 (102.8)	25.7 (102.8)
Tea with sugar (mL/day)	0 (0)	0 (0)	0 (0)	0 (0)	0 (0)	0 (0)	0 (0)	0 (0)	0 (0)	0 (0)
Soda (mL/day)	222.8 (445.7)	154.2 (480)	180 (471.4)	257.1 (411.4)	257.1 (360)	154.2 (342.8)	257.1 (385.7)	**281.8 (334.2)**	248.5 (445.7)	205.7 (411.4)
Flavored water beverages and home-made fruit beverages (mL/day)	0 (222.8)	0 (137.1)	0 (222.9)	0 (338.5)	0 (514.3)	0 (137.1)	68.6 (514.2)	85.7 (582.8)	0 (240)	51.4 (205.7)

Bold indicates statistically significant difference; *p* < 0.05 (Kolmogorov Smirnov test).

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
