# Peer review of "Dietary Sources of Fructose and Its Association with Fatty Liver in Mexican Young Adults"

_nutrients, 2019, doi:10.3390/nu11030522_

Round 1
Reviewer 1 Report
This is a report, written by Cantoral et al, of an observational study on potential contribution of fructose to development of fatty liver disease. Many previous reports have dealt with this issue, and the hypothesis has been gradually being accepted. The present study can be considered one of validation examinations, but seems to be inferior to preceding similar studies. For example, the authors should have demonstrated stronger points of their own study compared with a recent report published by Weber et al (Nutrients 2018; 10: 774). If I am a member of this study group, I propose collecting more patients with MRI-proven NAFLD, to show firmer evidence of association between fructose and NAFLD.
Author Response
This is a report, written by Cantoral et al, of an observational study on the potential contribution of fructose to development of fatty liver disease. Many previous reports have dealt with this issue, and the hypothesis has been gradually being accepted. The present study can be considered one of validation examinations, but seems to be inferior to preceding similar studies. For example, the authors should have demonstrated stronger points of their own study compared with a recent report published by Weber et al (Nutrients 2018; 10: 774). If I am a member of this study group, I propose collecting more patients with MRI-proven NAFLD, to show firmer evidence of association between fructose and NAFLD
Response: We agree that many previous studies had observed an association between dietary fructose intake and fatty liver disease. The recent study from Weber and colleagues is an important piece of evidence on how different sources of fructose (SSB, fruit) affect insulin sensitivity differently in patients with and without Diabetes; they also use FLI to diagnose steatosis in the sample. However, our study is one of the first in the Mexican population that includes 3 types of steatosis diagnostic methods (FLI, HSI, MRI- the gold standard method); also it is conducted in healthy (no previous diagnosed of DM) and relatively young population (21-22 years of age). Our study compares different diagnostic methods as well as observed differences in median intake of fructose according to the method of diagnosis. We also present the profile and diet characteristics of the false positives (positive identified by the index, but not diagnosed by the gold standard of MRI)
We have included the study of Weber, et al. in the discussion section to reference that a sample size around 100 has been reported previously to have enough power (80%) to detect associations between fructose intake groups and hepatic indexes. (Lines 326-329)
Reviewer 2 Report
This study examines dietary fructose consumption and the association with the patients by using three indexes: hepatic steatosis index (HSI), fatty liver index (FLI) and MRI diagnosed NAFLD in young adults from Mexico City. They found that sugar-sweetened beverages (SSB) consumption was significantly higher in patients with FLI ≥30, while natural fruit juices consumption was higher in NAFLD patients diagnosed by MRI. Moreover, they found SSB consumption is higher in false positive patients (FLI≥30, but not confirmed by MRI) which may have a predictive role in the development of NAFLD.
Overall, the paper is well written. However, several concerns need to be addressed.
SSB consumption was higher in patients with HSI≥36 and FLI≥30, respectively. The differences were statistically significant in patients with FLI≥30. However, SSB consumption was not significantly changed in NAFLD patients diagnosed by MRI, which should be included in most of patients with HSI≥36 and FLI≥30. On the contrary, NAFLD patients consumed more natural fruit juices whereas its consumption was zero in patients with HSI≥36 and FLI≥30. Presumably, NAFLD patients would consume SSB in a similar trend to the HSI and FLI positive patients. The discrepancy should be discussed. In addition, the role of natural fruit juices may be different from crystal fructose.
Do the authors have the follow-up data from the false positive patients with higher SSB consumption?
Knowledge on fructose metabolism need to be updated in the discussion (line 298-300). Fructose is primarily metabolized in the small intestine and spills to the liver only when consumed excess (Jang C, et al. Cell Metabolism 2018; 27:351-361).
“Hepatic steatosis and NAFLD” in the title seem to be conceptually overlapped. It would be better to reword it.
Several type errors: “HIS” in the title of Table 2; “FLI (>30)” in the first row of Table 3; “novo lipogenesis” in line 303 and 309; “leptine” in line 305.
Author Response
This study examines dietary fructose consumption and the association with the patients by using three indexes: hepatic steatosis index (HSI), fatty liver index (FLI) and MRI diagnosed NAFLD in young adults from Mexico City. They found that sugar-sweetened beverages (SSB) consumption was significantly higher in patients with FLI ≥30, while natural fruit juices consumption was higher in NAFLD patients diagnosed by MRI. Moreover, they found SSB consumption is higher in false positive patients (FLI≥30, but not confirmed by MRI) which may have a predictive role in the development of NAFLD.
Overall, the paper is well written. However, several concerns need to be addressed.
SSB consumption was higher in patients with HSI≥36 and FLI≥30, respectively. The differences were statistically significant in patients with FLI≥30. However, SSB consumption was not significantly changed in NAFLD patients diagnosed by MRI, which should be included in most of patients with HSI≥36 and FLI≥30. On the contrary, NAFLD patients consumed more natural fruit juices whereas its consumption was zero in patients with HSI≥36 and FLI≥30. Presumably, NAFLD patients would consume SSB in a similar trend to the HSI and FLI positive patients. The discrepancy should be discussed. In addition, the role of natural fruit juices may be different from crystal fructose.
We included in the discussion section that the discrepancy in SSB intake may be due to under-reporting in overweight and obese participants (prevalence of overweight in the sample are: 94% NAFDL, HSI 80%, FLI 65%, healthy 2%) as this previously have been reported. (Lines 296-300)
Do the authors have follow-up data from false positive patients with higher SSB consumption?
In February 2019, we started a new wave of data collection within these same participants and will collect new information and conduct additional analyses in the future.
Knowledge on fructose metabolism need to be updated in the discussion (line 298-300). Fructose is primarily metabolized in the small intestine and spills to the liver only when consumed excess
Thank you to the reviewer for the updated information. We incorporated Jang C, et al. study in our discussion. (Jang C, et al. Cell Metabolism 2018; 27:351-361). (Lines 307-309)
“Hepatic steatosis and NAFLD” in the title seem to be conceptually overlapped. It would be better to reword it.
We agree with the reviewer and have modified the title: “Dietary sources of fructose and its association with fatty liver in Mexican young adults”
Several type errors: “HIS” in the title of Table 2; “FLI (>30)” in the first row of Table 3; “novo lipogenesis” in line 303 and 309; “leptine” in line 305.
Thank you for pointing out the errors which have been corrected.
Reviewer 3 Report
Cantoral et al: Hepatic steatosis and NAFLD – Nutrients MDPI
Overall this is a very nicely written manuscript. I was surprised and concerned to find that NAFLD is prevalent in up to 20-30% of the Mexican population. Clearly finding cause-effect is important clinically and makes the manuscript a significant contribution. I think the manuscript will find excellent readership and should be accepted. As an editor I do not work for the beverage industry and do not have grants from the beverage industry, however I do feel the authors focus on “fructose” is detrimental to the main finding of the paper, that SSB is clearly predictive of NAFLD.
Title:
To me your title would be less controversial if it was “SSB intake association with hepatic…”, this modification would expose you to less negative bias and let the reader make that association less forcefully. Folks tend to be either anti-fructose or fructose-neutral, never in between, making it difficult for some to simple read the facts.
Abstract:
Nicely written and provided an excellent synopsis of the purpose, method and results of your study.
Introduction:
Very nice introduction to the background information and methods needed to understand your paper.
Line: 55 ….introduction as a sweetener in “THE” 1960s, due…..
Line 60-65: OK I get that the authors are anti-fructose, they also need to be fair and point out total calorie intake value from sugar, not just that 26% of intake is discretionary foods. Perhaps narrow this to just fructose intake only after you introduce the total sugar intake. If there are no studies looking at SSB total calorie intake in Mexico, perhaps they could use US adolescent intake figures and a benchmark for Mexico. You make also wish to point out the recent use of sugar taxes as a way of controlling intake, and stemming the tide of what could be an epidemic of weight control, diabetes and from what your paper suggests NAFLD: https://journals.plos.org/plosone/article?id=10.1371/journal.pone.0199337
Materials and Methods:
Thank you for a very thorough methods description, the authors did an excellent job of explaining what they did and why. I have only one question and that pertains to line 81. With regards to the n=100, I think this is saying that your request to meet was sent to 100 people “randomly?” chosen from the larger original 631 person cohort, and that each and every person sent a request ofr the weekend lab visit was adherent with the request. This seems odd and would suggest 100% adherence to the request. Clarifying this would help the paper significantly.
Typically, I usually see the ethics portion place toward the beginning of the methods, because the protocol review was completed prior to study data collection, perhaps consider moving this part to line 73?
Statistical description was thorough and seemed appropriate.
Results:
Very well written and few comments need be made.
Line 205: …. Consider: There were no statistically significant differences among these variables.
Line 210: you may wish to name the “two” variables to help the reader follow your paper with less effort.
Discussion:
Nicely written, I might again suggest less “fructose” vilification and more discussion of the problem related to total caloric intake. You may also wish to ponder mentioning SSBs contain carbonation and phosphates that can also contribute to metabolic changes. Overall however, I like your discussion and think it is appropriate.
Author Response
Overall this is a very nicely written manuscript. I was surprised and concerned to find that NAFLD is prevalent in up to 20-30% of the Mexican population. Clearly finding cause-effect is important clinically and makes the manuscript a significant contribution. I think the manuscript will find excellent readership and should be accepted. As an editor I do not work for the beverage industry and do not have grants from the beverage industry, however I do feel the authors focus on “fructose” is detrimental to the main finding of the paper, that SSB is clearly predictive of NAFLD.
Title:
To me your title would be less controversial if it was “SSB intake association with hepatic…”, this modification would expose you to less negative bias and let the reader make that association less forcefully. Folks tend to be either anti-fructose or fructose-neutral, never in between, making it difficult for some to simple read the facts.
We reconsider to modify the title as the suggestion of reviewer #2: “Dietary sources of fructose and its association with fatty liver in Mexican young adults”
Abstract:
Nicely written and provided an excellent synopsis of the purpose, method and results of your study.
Introduction:
Very nice introduction to the background information and methods needed to understand your paper.
Line: 55 …introduction as a sweetener in “THE” 1960s, due…..
We thank the reviewer for the observation and have corrected the sentence.
Line 60-65: OK I get that the authors are anti-fructose, they also need to be fair and point out total calorie intake value from sugar, not just that 26% of intake is discretionary foods. Perhaps narrow this to just fructose intake only after you introduce the total sugar intake. If there are no studies looking at SSB total calorie intake in Mexico, perhaps they could use US adolescent intake figures and a benchmark for Mexico. You make also wish to point out the recent use of sugar taxes as a way of controlling intake, and stemming the tide of what could be an epidemic of weight control, diabetes and from what your paper suggests NAFLD:
We included Sanchez T., et al paper about added sugar from SSB contribution in Mexican diet (lines 67-68).
Materials and Methods:
Thank you for a very thorough methods description, the authors did an excellent job of explaining what they did and why. I have only one question and that pertains to line 81. With regards to the n=100, I think this is saying that your request to meet was sent to 100 people “randomly?” chosen from the larger original 631 person cohort, and that each and every person sent a request ofr the weekend lab visit was adherent with the request. This seems odd and would suggest 100% adherence to the request. Clarifying this would help the paper significantly.
We specified in line 82 that the subsample of 100 participants was not randomly selected, but included as a way to compare internal validity with the original cohort. We have added supplementary table 1 that shows that were no differences between the original cohort and the subsample.
Regarding the adherence to the request: we tried to contact 206 young adults to participate in our study; of those, 55 did not answer and 51 did not agree to participate. This corresponds to a response rate of 50%.
Typically, I usually see the ethics portion place toward the beginning of the methods, because the protocol review was completed prior to study data collection, perhaps consider moving this part to line 73?
We have moved the ethical considerations to the beginning of methods section (lines 75-76)
Statistical description was thorough and seemed appropriate.
Results:
Very well written and few comments need be made.
Line 205: …. Consider: There were no statistically significant differences among these variables.
We have corrected the sentence as suggested.
Line 210: you may wish to name the “two” variables to help the reader follow your paper with less effort.
We have included the names of the indexes as suggested.
Discussion:
Nicely written, I might again suggest less “fructose” vilification and more discussion of the problem related to total caloric intake. You may also wish to ponder mentioning SSBs contain carbonation and phosphates that can also contribute to metabolic changes. Overall however, I like your discussion and think it is appropriate.
We included a paragraph in the discussion section related to total calories intake and NAFDL. (lines 340-343). And in lines 304-306 we included other health effects of SSB components.
Round 2
Reviewer 1 Report
I realized the authors' efforts to have tried to improve the manuscript, but I couldn't find a superior point in the revised manuscript compared with the preceding report of Weber et al. I regret to make such a comment.
Author Response
Please check attached version
